# Identification of a Plasma Microrna Signature as Biomarker of Subaneurysmal Aortic Dilation in Patients with High Cardiovascular Risk

**DOI:** 10.3390/jcm9092783

**Published:** 2020-08-28

**Authors:** Ana Torres-Do Rego, María Barrientos, Adriana Ortega-Hernández, Javier Modrego, Rubén Gómez-Gordo, Luis A. Álvarez-Sala, Victoria Cachofeiro, Dulcenombre Gómez-Garre

**Affiliations:** 1Internal Medicine Service, HGU Gregorio Marañón, Instituto Investigación Sanitaria Gregorio Marañón (IiSGM), 28007 Madrid, Spain; a.torres.do.rego@gmail.com (A.T.-D.R.); maria.barrientos@salud.madrid.org (M.B.); luisantonio.alvarezsala@salud.madrid.org (L.A.Á.-S.); 2Vascular Biology Research Laboratory, Hospital Clínico San Carlos-Instituto de Investigación Sanitaria del Hospital Clínico San Carlos (IdISSC), 28040 Madrid, Spain; a.ortega.hernandez@hotmail.com (A.O.-H.); javier.modrego@salud.madrid.org (J.M.); ruben.gomezgordo@gmail.com (R.G.-G.); 3Biomedical Research Networking Center in Cardiovascular Diseases (CIBERCV), 28029 Madrid, Spain; vcara@ucm.es; 4Department of Medicine, School of Medicine, Universidad Complutense, 28040 Madrid, Spain; 5Department of Physiology, School of Medicine, Universidad Complutense and Instituto de Investigación Sanitaria Gregorio Marañón (IiSGM), 28040 Madrid, Spain

**Keywords:** abdominal aortic aneurysm, early detection, microRNAs, circulating biomarkers

## Abstract

Patients with subaneurysmal aortic dilation (SAD; 25–29 mm diameter) are likely to progress to true abdominal aortic aneurysm (AAA). Despite these patients having a higher risk of all-cause mortality than subjects with aortic size <24 mm, early diagnostic biomarkers are lacking. MicroRNAs (miRs) are well-recognized potential biomarkers due to their differential expression in different tissues and their stability in blood. We have investigated whether a plasma miRs profile could identify the presence of SAD in high cardiovascular risk patients. Using qRT-PCR arrays in plasma samples, we determined miRs differentially expressed between SAD patients and patients with normal aortic diameter. We then selected 12 miRs to be investigated as biomarkers by construction of ROC curves. A total of 82 significantly differentially expressed miRs were found by qPCR array, and 12 were validated by qRT-PCR. ROC curve analyses showed that seven selected miRs (miR-28-3p, miR-29a-3p, miR-93-3p, miR-150-5p, miR-338-3p, miR-339-3p, and miR-378a-3p) could be valuable biomarkers for distinguishing SAD patients. MiR-339-3p showed the best sensitivity and specificity, even after combination with other miRs. Decreased miR-339-3p expression was associated with increased aortic abdominal diameter. MiR-339-3p, alone or in combination with other miRs, could be used for SAD screening in high cardiovascular risk patients, helping to the early diagnosis of asymptomatic AAA.

## 1. Introduction

Aneurysm is a pathological dilation that occurs mainly in the large arteries and most frequently in the abdominal portion of the aorta. Abdominal aortic aneurysm (AAA) usually develops asymptomatically and unpredictably in some cases, until its rupture. A global mortality of 80% is associated with this event [1]. Of these deaths, 30–50% of patients die before arrival at the hospital.

It has been shown that early detection of AAA reduces mortality in 42–66% of asymptomatic individuals [2]. Thus, screening of AAA is recommended in some countries for men aged between 60 and 75 years, especially if they have a history of smoking [3,4]. Although AAA is characterized by progressive expansion, an aortic diameter higher than 30 mm is commonly considered as aneurysmal. Thus, most AAA screening programs currently discharge patients with aortic diameters less than 30 mm [4], considering that these patients are not at risk of aortic rupture. However, mild aortic dilations of 25.0 to 29.9 mm, also known as aortic ectasia, could have clinical consequences [5,6]. Several studies have shown that patients with ectatic abdominal aorta show high cardiovascular risk [7,8]. Importantly, these patients have a higher risk of all-cause mortality with respect to subjects with aortic size <24 mm [9]. In addition, both observational and retrospective studies have demonstrated that patients with ectatic aortas are likely to progress and develop an AAA [10,11]. Therefore, the identification of ectatic aortas/subaneurysmal aortic dilatations (SAD) could be considered, not only a parameter of early identification of patients with asymptomatic AAA but also a marker of subclinical end organ damage that could be related to an increased risk of cardiovascular complications.

Ultrasonography (US) is presently the method of choice for AAA screening due to its simplicity, validity, and cost-effectiveness [12]. Although it can be performed by operators after a short period of specific training [13], the need to perform the procedure at often-overburdened ambulatory clinics or hospitals could limit its use in large-scale population screening programs. Thus, the identification of serologic diagnostic markers could be a useful complementary tool for clinical management.

MicroRNAs (miRNAs or miRs) are single-stranded small noncoding RNA molecules of 20–22 nucleotides in length that have been shown to be important negative regulators of gene expression by binding to target messenger RNAs (mRNA) and inhibiting their translation or promoting their degradation [14]. However, miRs can also up-regulate gene expression by inhibiting the expression of transcription regulators [14]. In the human genome, approximately 2600 miRs have been identified that can regulate almost 60% of human protein coding genes [14]. Importantly, it is estimated that each miR targets several mRNAs and, at the same time, unique mRNA can be regulated by several miRs [14]. Since regulated miR expression has been shown to be essential for maintaining cellular homeostasis, alterations in miR expression profiles have been associated to many diseases, including cardiovascular disease (CVD) [15]. Regarding AAA, miRs have been consistently down- or upregulated in aortic tissue and blood samples from patients with AAA compared with controls in several studies [16,17]. Along these lines, miRs can modulate the inflammatory and remodeling processes associated with aneurysm development [16]. In addition to the differential expression of miRs observed in some tissues at pathophysiological conditions, miRs can be specifically released into extracellular biofluids, including blood. Since circulating miRs can be found in vesicles or associated with proteins, they are highly stable despite multiple freeze-thaw cycles, boiling, or extreme pH [18] and can be accurately assayed by sequence-specific techniques [19]. Thus, there is increasing evidence that the study of miRs could generate new biomarkers. In this sense, some miRs have been proposed as suitable biomarkers for CVD in general [15] and for AAA in particular [20]. Interestingly, analyses of a combination of several miRs and clinical parameters appear to achieve higher accuracy in AAA diagnoses [21,22]. However, the expression of circulating miRs in patients with SAD has not yet been reported.

Hence, in this study, we have aimed to identify and validate a profile of plasma miRs that could be useful for early AAA detection. For this purpose, we have analyzed the plasma miR levels from patients with ectatic abdominal aorta/SAD, since it has been reported that these patients are likely to progress and develop an AAA.

## 2. Experimental Section

### 2.1. Study Design and Population

The current study was performed with a cohort of patients over 50 years of age with high or very high cardiovascular risk according to European Society of Cardiology and other societies on cardiovascular disease prevention in clinical practice [23] who attended the Internal Medicine Service of the Hospital Gregorio Marañón for AAA screening for one year following a protocol previously reported [8]. Upon enrollment, detailed information on family and personal medical history (including smoking and alcohol habits) and regular medication was recorded, anthropometric measurements (weight, height, and waist circumference) were assessed, blood pressure was taken after 15 min rest in decubitus by an oscillometric device (Microlife Watch BP office ABI, Widnau, Switzerland), and a fasting blood sample was obtained for biochemical analyses. Plasma and serum samples were stored at −80 °C for future studies.

The measurement of the infrarenal abdominal aorta diameter was performed using a convex probe (CA631, MyLab Twice, Esaote, Barcelona, Spain) by an electronic caliper. Measurements were made with the patient in supine, or in left lateral decubitus position when necessary, after 15 min of rest. The aortic diameter measurement was performed at the point of greatest dilation from the renal arteries to the bifurcation of the iliac arteries. It consisted of the mean of 2 measurements of the maximum anteroposterior diameter in a longitudinal plane and the mean of 2 measurements of the maximum transverse diameter in a coronal plane, taking as reference the external wall of the aorta in systole. All measurements were made on frozen images by the same investigator. Based on abdominal ultrasonography results, participants were categorized as the control group (patients with abdominal aorta diameter less than 25 mm), the SAD group (patients with aortic size from 25.0 to 29.9 mm), and the AAA group (patients with abdominal aorta ≥30 mm).

The protocol of this study complies with the principles of the Helsinki Declaration and Good Clinical Practice Guidelines and was approved by the Clinical Research Ethics Committee of Gregorio Marañón Hospital (Madrid, Spain). Each participant singed a written informed consent document.

### 2.2. Sample Collection

Peripheral blood was collected in 10 mL tubes containing ethylenediaminetetraacetic acid (EDTA) as an anticoagulant. Tubes were centrifuged at 1500× *g* for 10 min at 4 °C. Plasma was recovered, dispensed in aliquots of 500 µL, and stored at −80 °C until used.

### 2.3. RNA Isolation

We excluded blood samples from patients with clinical and/or laboratory evidence of recent infection, previous cardiovascular event in the last 3 months, cancer, autoimmune disease, or terminal disease within the 3 groups, as well as from AAA patients submitted to aneurysm repair surgery. In addition, in order to avoid misunderstandings with respect to the results obtained with patients with SAD, an age- and time-of-hypertension-matched group of 32 patients with abdominal aortic diameter less than 25 mm was analyzed as control. Then, a plasma aliquot from 32 patients with abdominal aorta diameter less than 25 mm, 30 patients with aortic size from 25.0 to 29.9 mm, and 15 patients with abdominal aorta ≥30 mm was thawed on ice and centrifuged at 1500× *g* for 10 min at 4 °C to remove debris. Total RNA isolation was performed using the Serum and Plasma miRNeasy kit (Qiagen, Hilden, Germany), according to the manufacturer’s instructions. RNA quality control was conducted using 260/280 ratios, and only samples with a ratio between 1.8 and 2.2 were included. Three synthetic RNA spike-ins at various concentrations (UniSp2, UniSp4, and UniSp5) were used to check for RNA isolation efficiency.

### 2.4. cDNA Synthesis and Identification of Plasma miR Expression Profiling

For the screening experiment, isolated miR plasma samples from 6 controls, 6 patients with SAD, and 6 patients with AAA were pooled in their respective groups. This approach is based on previous studies from our group [24]. For each pool, 40 ng of total RNA was employed for cDNA synthesis with the miR CURYLNA Universal RT microRNA PCR System (Exiqon, Vedbaek, Denmark), following the manual’s instructions. Reverse transcription reaction efficiency and polymerase chain reaction (PCR) inhibitor presence was checked with UniSp6 and cel-miR-39-3p, respectively. Prepared cDNA samples were stored at −20 °C until PCR determinations were performed using the miR CURYLNA Universal RT microRNA PCR, Human panel I + II (Exiqon) following the manual’s instructions. This panel allows for the analysis of 752 human miRs. The amplification was performed using an ABI 7900HT qPCR instrument (Thermo Fisher Scientific, Carlsbad, CA, USA) in 384-well plates. The amplification curves were analyzed using SDS v.2.4 software (Thermo Fisher Scientific) for determination of threshold cycles (Ct). In order to identify plasma expression miR profiling, an initial selection from the 752 human miRs analyzed was performed by eliminating 410 miRs, given they were not expressed in any group. All values showing as “undetermined” were then replaced by 35 Ct, and data were normalized using the global mean of all miRs with Ct values <35 (ΔCt) as a reference control [25]. Fold changes were then calculated using the 2^-∆∆Ct method. Twelve miRs were selected for validation by quantitative real-time (qRT)-PCR.

### 2.5. Validation of Plasma miR Expression Profiling by qRT-PCR Analysis

To investigate whether any of the miRs could be associated to the presence of ectatic aorta/SAD, differential expression of the 12 selected miRs (let-7b-5p, let-7e-5p, miR-27a-3p, miR-28-3p, miR-29a-3p, miR-93-3p, miR-133b, miR-150-5p, miR-331-3p, miR-338-3p, miR-339-3p, and miR-378a-3p) was individually validated by qRT-PCR in 32 controls and in 30 patients with SAD and 15 with AAA, including the individuals used to create the pools. Thus, 10 ng of total RNA from each individual sample was employed for cDNA synthesis as described above, and qRT-PCR was then performed for each miR of interest using SYBR Green master mix and specific LNA PCR primer sets (Exiqon), following the manufacturer’s instructions. We tested five candidates to reference gene, which were selected among genes that may be stably expressed in plasma/serum samples based on the literature or pre-existing data (Appendix A). MiR-451a showed to be good candidate to be a reference gene for normalization.

### 2.6. Target Gene Prediction and Functional Enrichment Analysis

Target gene prediction and functional enrichment analysis of miR-339-3p were performed using miRNet (http://www.mirnet.ca/) [26]. Unbiased empirical sampling was the statistical analysis used to infer the null distribution of the target genes as selected based on the input miRNAs. The protein annotation through evolutionary relationship (PANTHER) classification system pathway enrichment analysis was subsequently performed for the target genes (http://www.pantherdb.org/) [27].

### 2.7. Statistical Analysis

Categorical variables are expressed as frequencies and percentages and were analyzed by the chi-square test or by the Fisher’s exact test in case more than 25% of the expected values were less than five. Normal distribution of quantitative variables was assessed with the Kolmogorov–Smirnov test. For variables with normal distribution, data are expressed as media ± Standard Deviation (SD) and were analyzed with the ANOVA test with post hoc Bonferroni correction for multiple comparisons after assessing homogeneity of variances with the Levene test. For non-normal distributed variables (triglycerides, glucose and miRs), data are expressed as median and interquartile range and were analyzed using the Kruskal-Wallis test with post hoc Mann-Whitney U-test. We used the corrected *p*-value for multiple corrections. The diagnostic value, including sensitivity and specificity, of selected miRs was investigated by generating the receiver operating characteristic (ROC) curve and calculating the area under the curve (AUC). Spearman’s Rho correlation coefficients were calculated to examine linear relationships between 12 miRNAs and aortic diameter. The level of significance was defined as a *p*-value < 0.05 divided by the number of miRs (0.0042). A multivariate linear regression analysis was fitted in order to evaluate the variables associated with the expression levels of miRs. These analyses were performed after adjustment for sex, age, and smoking status. All data were analyzed using SPSS software version 22.0 (SPSS Inc, Chicago, IL, USA), and a *p*-value < 0.05 was considered statistically significant.

## 3. Results

### 3.1. Clinical Characteristics of Patients

A total of 300 patients with high cardiovascular risk, with a median age of 67 years and 54% males were included. Demographic, clinical, and laboratory characteristics of the patients according to abdominal aortic size are shown in Table 1. In our cohort, we detected a total of 47 patients (15.7%) with SAD and 22 (7.3%) with AAA. Patients with SAD were predominantly male, a 66% with a history of smoking, all patients had arterial hypertension and 66% dyslipidemia. Diabetes mellitus was present in only 23% of patients. Some patients already showed coronary heart disease (CHD), cerebrovascular disease (CVD), and peripheral artery disease (PAD). This profile was like that of patients with established AAA although the latter were older. Our analysis showed that 12 patients (40%) with SAD and 3 (20%) with AAA were within the age group of 50 to 64 years old.

### 3.2. Identification of Plasma miRs Differentially Expressed in Patients with SAD

First, we performed a screening experiment of miRs detectable in blood samples from our patients by profiling pooled plasma samples from controls and patients with SAD and AAA with a miR qPCR-array experiment. For each pool, we selected consecutive six first patients included in each group taking into account sex distribution. Demographic and clinical characteristics of these patients are shown in Appendix A.

Of the 752 miRs analyzed, 342 miRs were detected with Ct < 35 in at least one of the studied groups: 314 miRs were present in the pool of controls, 220 in the patients with SAD, and 198 in the patients with AAA (Figure 1a), displaying average Ct values 30.0, 29.5, and 31.2, respectively. To determine differentially altered miR levels in the patients with SAD, miRs ≥ 2-fold (corresponding to a 1 cycle difference in qPCR assays) increased or decreased with respect to controls were identified (Figure 1b). Most miRs ≥2-fold changed were decreased (85.4%) in the SAD group.

### 3.3. Validation of miR Expression by Individual qRT-PCR

Given that the presence of SAD was associated with a diminution in the number of detected miRs, we selected 12 miRs with ≥2-fold reduced expression in patients with SAD in comparison with controls. Six miRs were selected since they had been previously associated to aorta aneurysms (let-7b-5p, miR-27a-3p, miR-29a-3p, miR-93-3p, miR-133b, miR-331-3p). Let-7e-5p, miR-150-5p, and miR-378a-3p had been previously related to molecular mechanisms associated with the development of AAA disease, and miR-28-3p, miR-338-3p, miR-339-3p had the great fold decreased. In agreement with data obtained in the screening experiment, selected miRNAs were markedly downregulated in patients with ectatic aorta/SAD compared to controls (Figure 2). Interesting, some of them were further decreased in patients with AAA (Figure 2).

### 3.4. Plasma miRs as Predictors of SAD in Patients with High Cardiovascular Risk

To investigate the diagnostic potential of miRs to discriminate between patients with normal aortic diameter and those with SAD, ROC curves were performed for each of the 12 miRs. When assessing single miRs, moderate AUC values (>0.7) were attributed to most of the miRs selected, although the expression of miR-339-3p had the highest performance (Table 2).

Therefore, we also tested whether combining the values of miR-339-3p with all the differentially expressed miRs led to even better performance; however, none of the combinations resulted in superior discriminatory capability compared with miR-339-3p alone. Even the combination of several miRs did not show higher AUC values than miR-339-3p alone (Figure 3a); the highest achieved AUC value, obtained with the combination of miR-339-3p + let-7b-5p + miR-150-5p, is shown in Figure 3b. Indeed, by using the cutoff point 0.548, the sensitivity and specificity of miR-339-3p to differentiate patients with SAD from controls were 80.77% (95% confidence interval (CI) 69.62–91.92%) and 77.27% (95% CI 65.42–89.13%), respectively. None of the studied miRs achieved higher specificity with the same sensitivity (80%) (Table 2). A multivariate logistic regression analysis was also performed, showing that miR-339-3p in plasma can be a potential diagnostic biomarker for the identification of patients with SAD after adjustment for age, sex, and smoking status (odds ratio (OR) (95% CI) 8.78 (1.86–41.45); *p* = 0.006).

### 3.5. Correlations of miRs with Clinical Parameters

There was a strong negative correlation between almost all miRs and the aortic diameter (Table 3), including miR-339-3p (Figure 4). However, only miR-150-5p and miR-29a-3p negatively correlated with age (r = −0.353, *p* = 0.005 and r = −0.341, *p* = 0.008, respectively). MiR-93-3p negatively correlated with systolic blood pressure (r = −0.272, *p* = 0.039). There was no correlation between other biochemical parameters and the differentially expressed miRs.

### 3.6. Target Gene Prediction and Functional Enrichment Analysis

To investigate the potential biological activity of miR-339-3p, we predicted its target genes using miRNet. Fourteen genes were targeted by miR-339-3p: ATP5, KLHL15, WDTC1, PIPNM3, MEX3D, NFKB1, MCL1, FOXO1, PHLDA2, IGF2, RPL10A, SHOX2, AKIRIN1, and HEYL. PANTHER pathway enrichment analysis of these target genes revealed that they were involved in various pathways related to AAA (Figure 5).

## 4. Discussion

Our data demonstrate that patients with ectatic aorta/SAD could be identified with a panel of plasma circulating miRs, including let-7b-5p, let-7e-5p, miR-27a-3p, miR-28-3p, miR-29a-3p, miR-93-3p, miR-133b, miR-150-5p, miR-331-3p, miR-338-3p, miR-339-3p, and miR-378a-3p, which were differently expressed in patients with SAD with respect to patients without abdominal aortic dilation (diameter < 25 mm). Among them, miR-339-3p was demonstrated to be a potent biomarker of SAD presence even after controlling for important confounders, such as age, sex and smoking status. To our knowledge, we are the first to identify the association of miR plasma levels with the presence of ectatic aorta/SAD.

MiRs have emerged as regulatory molecules of gene expression and have been shown to play a key role in vascular biology and cardiovascular diseases [14]. In this context, several studies have identified the involvement of miRs in AAA formation and complications, as well as biomarkers of AAA diagnosis, although with divergent results that could be partially explained by the origin of the samples [15,16,17]. Most studies have used aortic tissue, and it has been reported that the expression of miRs can differ along the aorta, depending on its grade of dilation [28].

In addition, aortic tissue can only be obtained from patients at the time of the surgical intervention. The discovery that miRs can be detected in body fluid and that they are very stable led to their proposal as noninvasive biomarkers in many diseases [11]. Several studies have reported a differential miR expression in plasma and/or in serum between patients with AAA and patients without aortic dilations [16,17,20,21,22]. Interestingly, most miRs upregulated in aortic tissue have been found to be downregulated in circulating blood in patients with AAA [29], proposing that changes in the aortic wall could be reflected in the circulation. However, the role of circulating miRs as diagnostic markers has not been established.

All these studies had investigated patients already diagnosed with AAA. Recent evidence indicates that most patients with SAD progress to true AAA [9,11]. Our data demonstrate that some decreased miRs in AAA patients were already reduced in SAD patients in comparison with the levels of these miRs in patients without abnormal aortic dilation. Aneurysms are characterized by endothelial dysfunction, chronic inflammation, extracellular matrix degradation, and gradual thinning of the vascular wall due to apoptosis of vascular smooth muscle cells (VSMCs) [30]. Most miRs analyzed in our study were previously related to molecular mechanisms associated with the development of AAA disease. There is strong evidence that the miR-29 family is a regulator of collagen synthesis and fibrosis both in vitro and in vivo [31]. The addition of a miR-29a precursor in VSMCs was associated with a reduced expression of MMP-2, a metalloprotease shown to have increased expression in aortic aneurysms [32]. More recently, the downregulation of miR-29a-3p has been reported to reduce the cell viability of AAA cells [33]. MiR-331-3p and miR-133b have been reported to regulate inflammation and apoptosis [34,35]. Several studies have reported the important role of miRs such as miR-27, miR-378, miR-150, and the let-7 family, specifically let-7b, in regulating endothelial cells angiogenesis [34,35,36]. In addition, various studies have demonstrated that the let-7 family is downregulated in patients with cranial aneurysm [37] and AAA [20,21], suggesting its participation in common pathways linked to different forms of aneurysm. MiR-150 has been associated with AAA due to the ability of certain stimuli associated with the progression and formation of AAA, such as oxidative stress or an inflammatory state, to induce its release by monocytes in vitro [22]. It has recently been suggested that miR-150 could be involved in the pathophysiology of AAA by playing a protective role. MiR-150 regulated endothelial cell function and vascular remodeling in an experimental model by inhibiting pentaxin-3, whose expression in aortic tissue has been negatively correlated with maximum aortic diameter in patients with AAA [38,39].

Among the miRs studied, miR-339-3p showed the largest discriminatory power to identify patients with SAD. No studies have previously associated miR-339-3p with AAA. MiR-339 is highly expressed in the cardiovascular system and an important regulator of VSMC proliferation in pulmonary arteries [40]. Predicted genes targeted by miR-339-3p have been involved in adipocytokine, NOD-like receptor, chemokine, insulin, and MAPK signaling pathways and in apoptosis, all of them implicated in AAA pathology [41,42]. In addition, miR-339-3p could be downregulated by many cytokines and growth factors [43], suggesting a mechanism by which it could be significantly reduced in patients with SAD. Additional studies in patients and animal models are needed to determine the precise role of miR-339-3p in SAD progression.

One of the major challenges in the prevention of AAA development is early detection of aortic dilation before an irreversible rupture occurs. However, in this study, we have found that patients with SAD showed similar incidence of CHD and PAD, and higher CVD than patients with true AAA, despite being younger. In a previous study, we have found that SAD patients already showed increased intima-media values as well as a greater presence of carotid plaques than patients without abdominal aortic dilation. In agreement with our results, it has been reported that men with aortic diameters > 25 mm (SAD) were at increased risk of hypertensive disease, ischemic heart disease, and chronic obstructive pulmonary disease, as well as an increased risk of diabetes mellitus and of heart failure compared with men with an aortic diameter of 24 mm or less [11]. In addition, the value of plasma D-dimer has been reported to be significantly higher in patients with SAD than in control subjects with a normal diameter of the abdominal aorta, possibly reflecting the presence of microthrombi in the wall of the aorta [44]. We believe that the importance of identifying SAD patients is not only as they could develop AAA in the future but also to identify patients with an increased risk of cardiovascular complications and all-causes of mortality.

Most screening programs are related to male gender, age ≥ 65 years, and smoking. However, in our study, 40% of patients with SAD and 20% with AAA were under 65 years [45]. These populations would never be identified with the screening programs, suggesting that targeted screening of AAA should be tailored to younger patients with high cardiovascular risk. For this purpose, the identification of serologic diagnostic markers could be a useful tool for large-scale screening programs outside the hospital but also a complement for assisting doctors in the clinical decision-making. Interesting, very recently Lu et al. have developed a portable interdigitated electrode (IDE) sensing surface to identify miRs with good reproducibility [46].

Our study has certain limitations. We have focused on downregulated miRs. However, we cannot exclude that some upregulated miRs could also act as biomarkers of SAD. Moreover, selected miRs are related to vascular processes of AAA development, such as inflammatory response, cardiac hypertrophy, angiogenesis, and fibrosis, although they are not exclusive for this pathology. In addition, all patients included in this study were hypertensive. Further studies are needed to support these results in the general population in order to improve screening programs.

## 5. Conclusions

Our data demonstrated a miR profile associated with SAD in high cardiovascular risk patients that could be used for early AAA identification. Plasma panels of miRs in association with the progressive advancement in nano-biosensing technology [47] could provide us with rapid, inexpensive miniaturized devices for precise screening and early diagnosis of AAA in the near future. Future studies should evaluate its usefulness as a biomarker of disease progression.

## Figures and Tables

**Figure 1 jcm-09-02783-f001:**
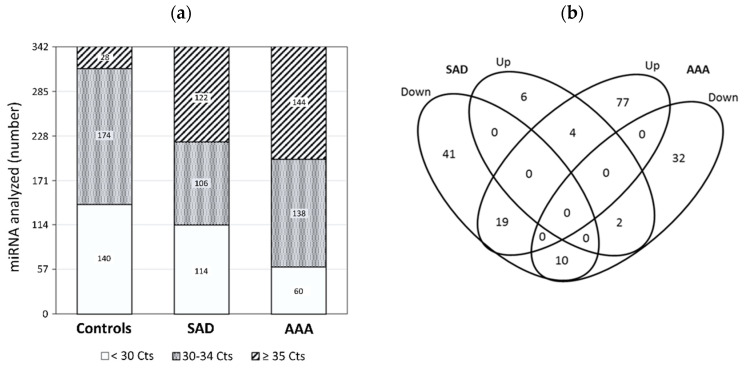
Identification of miRs expression by PCR (polymerase chain reaction) array: (**a**) Distribution of Ct values of 342 miRs expressed in controls, in patients with subaneurysmal aortic dilations (SAD), and in patients with aneurysmal aortic dilations (AAA); (**b**) Venn diagram of miR analysis in patients with SAD and AAA. The miRs ≥ 2-fold differentially increased or decreased in comparison with controls are depicted in 4 overlapping ellipses. The numbers indicate the miR counts in the indicated area. miRNA, MicroRNAs.

**Figure 2 jcm-09-02783-f002:**
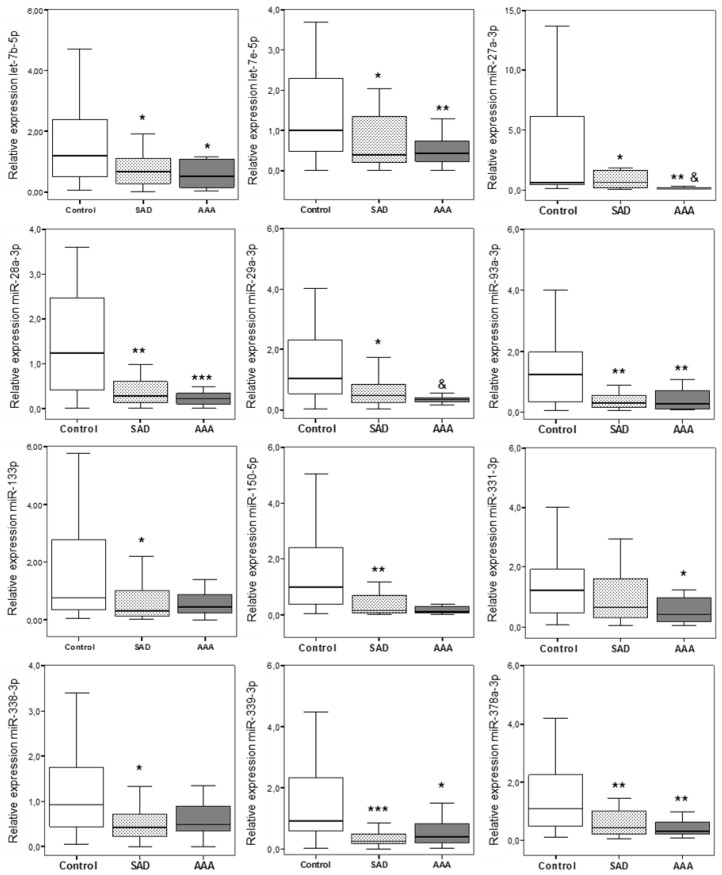
Plasma expression levels of let-7b-5p, let-7e-5p, miR-27a-3p, miR-28-3p, miR-29a-3p, miR-93-3p, miR-133b, miR-150-5p, miR-331-3p, miR-338-3p, miR-339-3p, and miR-378a-3p in controls (*n* = 32), and in patients with subaneurysmal aortic dilations (SAD) (*n* = 30), and patients with aneurysmal aortic dilations (AAA) (*n* = 15). MiRs were detected by qRT-PCR in plasma samples and calculated using the ΔCt method with miR-451 used as the internal normalizer. Data are presented as box plot with median and interquartile range. * *p* < 0.05 vs. controls; ** *p* < 0.01 vs. controls; *** *p* < 0.001 vs. controls and *p* < 0.01 vs. SAD. & *p* < 0.01 vs. SAD.

**Figure 3 jcm-09-02783-f003:**
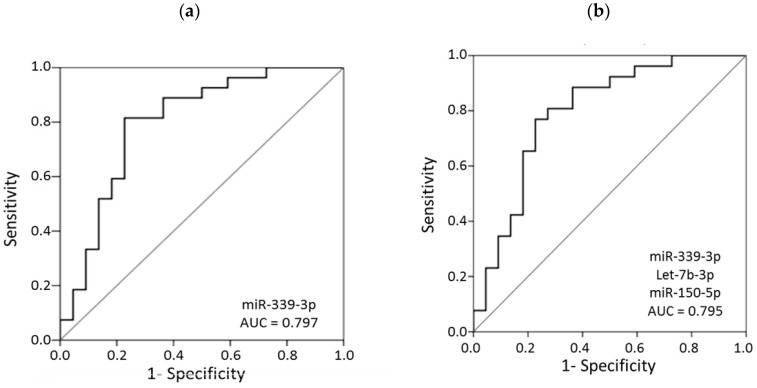
Receiver operating characteristic (ROC) curves for (**a**) miR-339-3p and (**b**) the combination of miR-339-3p + let-7b-5p + miR150-5p to discriminate patients with subaneurysmal aortic dilations from patients with normal aortic diameter. AUC, area under the curve.

**Figure 4 jcm-09-02783-f004:**
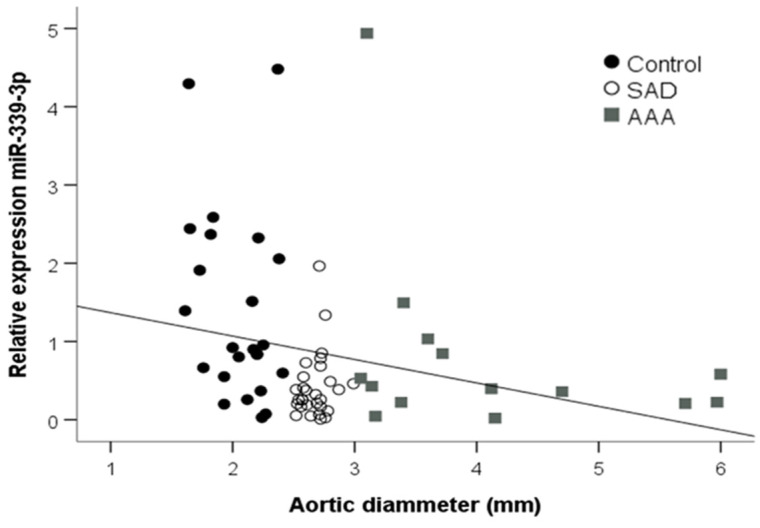
Linear correlations between the plasma miR-339-3p concentration and the aortic diameter of patients without aortic dilation, patients with subaneurysmal aortic dilations, and patients with aneurysmal aortic dilations. SAD, subaneurysmal aortic dilation; AAA, abdominal aortic aneurysm.

**Figure 5 jcm-09-02783-f005:**
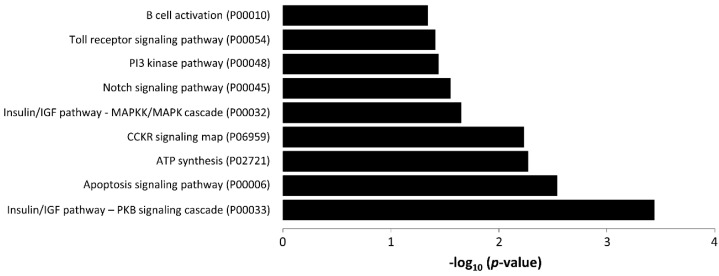
PANTHER analysis showing pathways related to hsa-mir-339-3p target genes (*p* ≥ 0.05). The negative log10 of the *p*-value is plotted on the *x*-axis.

**Table 1 jcm-09-02783-t001:** Clinical and biochemical features of patients with normal aortic diameter (CT), patients with ectatic aorta/subaneurysmal aortic dilations (SAD), and patients with aneurysmal aortic dilations (AAA).

	All	CT	SAD	AAA
	(*n* = 300)	(*n* = 231)	(*n* = 47)	(*n* = 22)
Age, years	67 ± 3	67 ± 3	66 ± 5	75 ± 3 *
Male, *n* (%)	162 (54.0)	105 (45.5)	40 (85.1) *	17 (77.3) *
BMI, kg/m^2^	28.3 ± 1.1	28.4 ± 1.0	28.7 ± 1.7	27.5 ± 2.3
WC, cm	98 ± 3	97 ± 3	101 ± 4	100 ± 4
Smoking habits, *n* (%)	131 (35.7)	85 (36.8)	31 (66.0) *	15 (68.2) *
Alcohol, *n* (%)	51 (17.1)	29 (12.6)	16 (34.0) *	6 (27.3) *
Hypertension, *n* (%)	300 (100)	231 (100)	47 (100)	22 (100)
SBP (mmHg)	140 ± 3	141 ± 3	140 ± 4	132 ± 7
DBP (mmHg)	80 ± 5	80 ± 3	80 ± 5	75 ± 4
Dyslipidemia, *n* (%)	178 (59.3)	134 (58.0)	31 (66.0)	13 (59.1)
Triglycerides, mg/dL	96 (74–134)	98 (74–129)	110 (71–156)	102 (80–142)
TC, mg/dL	188 ± 8	189 ± 8	182 ± 11	163 ± 13 *
LDL-c, mg/dL	111 ± 7	111 ± 7	111 ± 10	92 ± 14
HDL-c, mg/dL	54 ± 3	52 ± 3	53 ± 5	47 ± 6 *
T2DM, *n* (%)	66 (22.0)	50 (21.6)	11 (23.4)	5 (22.7)
Glucose, mg/dL	99 (90–111)	99 (89–111)	98 (94–116)	99 (90–119)
CHD, *n* (%)	19 (6.3)	9 (3.9)	6 (12.8) *	4 (18.2) *
CVD, *n* (%)	23 (7.7)	16 (6.9)	6 (12.8) *	1 (4.5)
PAD, *n* (%)	12 (4.0)	6 (2.6)	2 (4.3) *	4 (18.2) *
eGFR, ml/min/1.72 m^2^	84.0 ± 4.5	84.5 ± 4.3	86.0 ± 7.3	70.5 ± 7.5 *

Values are expressed as media ± SD or median (interquartile range) for continuous variables or *n* (%) for categorical variables. * *p* < 0.05 vs. CT group. Abbreviations: BMI, body mass index; WC, Waist circumference; SBP, systolic blood pressure; DBP, diastolic blood pressure; TC, Total cholesterol; LDL-c, low density lipoprotein cholesterol; HDL-c, high density lipoprotein cholesterol; T2DM, diabetes mellitus type 2; CHD, coronary heart disease; CVD, cerebrovascular disease; PAD, peripheral artery disease; eGFR, estimated glomerular filtration rate.

**Table 2 jcm-09-02783-t002:** Receiver operating characteristic (ROC) curves of single miRs to discriminate between SAD patients and control hypertensive patients.

	AUC	95% Confidence Interval	Specificity *	*p*-Value
miR-339-3p ^†^	0.797	0.665–0.930	77.3	<0.001
miR-28-3p	0.741	0.599–0.884	54.5	0.004
miR-93-3p	0.731	0.582–0.880	63.6	0.006
miR-378a-3p	0.731	0.589–0.873	40.9	0.006
miR-150-5p	0.729	0.583–0.875	45.5	0.007
miR-338-3p	0.713	0.560–0.867	54.5	0.012
miR-29a-3p	0.710	0.562–0.857	38.7	0.013
let-7b-5p	0.698	0.546–0.849	45.5	0.019
miR-133b	0.687	0.538–0.836	34.1	0.027
let-7e-5p	0.677	0.524–0.829	31.8	0.037
miR-331-3p	0.626	0.467–0.785	31.8	0.136
miR-27a-3p	0.542	0.377–0.707	27.3	0.619

* Specificity (%) reached at 80% sensitivity. ^†^ ROC curve for miR-339-3p is shown in Figure 3a. Abbreviations: AUC, Area under curve. *p*-value < 0.0042 was defined as statistically significant.

**Table 3 jcm-09-02783-t003:** Linear correlations between miRs and aortic diameter of patients without aortic dilation, patients with subaneurysmal aortic dilations, and with aneurysmal aortic dilations.

	r	*p*-Value
let-7b-5p	−0.172	0.184
let-7e-5p	−0.363	0.005
miR-27a-3p	−0.295	0.021
miR-28-3p	−0.392	0.002
miR-29a-3p	−0.519	<0.001
miR-93-3p	−0.354	0.005
miR-133b	−0.311	0.017
miR-150-5p	−0.477	<0.001
miR-331-3p	−0.282	0.030
miR-338-3p	−0.259	0.044
miR-339-3p	−0.345	<0.001
miR-378a-3p	−0.349	0.007

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
