# Peer review of "Identification of a Plasma Microrna Signature as Biomarker of Subaneurysmal Aortic Dilation in Patients with High Cardiovascular Risk"

_jcm, 2020, doi:10.3390/jcm9092783_

Round 1

Reviewer 1 Report

The manuscript aimed at assessing the utility of microRNAs (miRs) as potential biomarkers of abdominal sub-aneurysmal aortic dilation, a condition that can subsequently progress to abdominal aortic aneurysm.

ISSUES

  1. In the Statistical Analysis section, 2.6:
    1. the use of the term qualitative in the context of analysis using chi-square/Fisher’s exact is confusing. These analytical techniques use quantitative data, so it is not clear how qualitative data were analyzed using these techniques. Was the intent to say that qualitative data were converted to quantitative data that then allowed use of these methods?
    2. Spelling error, second sentence “Quantitative …as media…”. I assume this was meant to spell median.
    3. Mention is made of Bonferroni adjustments for multiple comparisons- where this was applied should be clearly indicated in the Results sections that follow; this means that the p-value used when there was multiple testing was not 0.05 but something smaller.
  2. The number of participants in each of the three groups (Control, SAD and AAA) needs be clarified as there are different places with different numbers indicated:
    1. On page 3, Section 2.3 it is indicated that miR plasma was isolated from 6 patients in each of these groups
    2. On page 4, Section 3.1, discusses 12 patients with SAD and 3 with AAA in the age group 50-64. How, if at all, do these numbers relate to the 6 per group mentioned above?
    3. On page 5, Section 3.2, there is mention of 32 patients with AAA less than 25 mm who were analyzed as a control. How does this relate to the sample size descriptions above?
  3. Section 3.2
    1. Time-of-hypertension was used to identify matched controls: is this time of diagnosis of hypertension? If so, how accurate is that since someone might have hypertension prior to an official diagnosis
  4. Some clarification as to why the 2-fold threshold was used in identifying the miRs.
  5. Figure 2:
    1. The n should be given for each of the three groups in each of the panel
    2. What is the meaning of the & sign?
  6. Section 3.4
    1. The OR and its 95%CI are rather large which is a function of the small sample size.
    2. Was smoking considered for adjustment?
  7. Figure 4:
    1. If possible, please demarcate on the figure (e.g. with a different shape of mark) the three groups as indicated in the legend of the figure.
  8. In the absence of a clear indication of a temporal association between miRs appearance and SAD (and AAA), conclusions/statements that are suggestive that miRs might need to be reconsidered as the design of the study does not allow for clearly establishing such a temporal relationship. Could the miRs down/upregulation be a result of SAD/AAA rather than a cause?

Reviewer 2 Report

Review

The authors evaluate plasma miRNA as biomarkers for SAD. They identify decreased miRNA-339-3p as a potential predictor for SAD. The rational for the study is reasonable, however there are a number of questions and suggestions the authors must address. 

In the text you refer to aneurysmal diameter between 25 cm to 29.9 as sub-aneurysmal. This is traditionally referred to as ectasia. Suggest changing to ectasia throughout the text.

A major drawback to this paper is the sample size. Unfortunately, a sample size of 6 in SAD and AAA is simply not enough to account for the differences between patients. The authors should perform a power calculation to get a better idea re: size. Most genomic based biomarker studies use sample sizes from 20-50. I understand it is difficult to perform a power calculation when there are so many different miRNAs that are being assessed.

The study can be strengthened by the addition of aortic tissue. In the discussion, the authors claim that plasma biomarkers in AAA disease are a reflection of aortic wall changes. Can the authors compare miRNA profiles (from previously published datasets) to the ones they found to be differentially expressed.

Specific comments:

Abstract

Please specify what are the clinical consequences.

Sub-aneurysmal dilation is traditionally referred to as “ectasia”. Suggest authors change the naming appropriately.

Introduction

  1. General - need to expand more on the role of miRNAs. What are they and how do they function? Why are they suitable biomarkers? What makes them stable in the blood?

  1. Need to have a stronger and well delineated objective sentence. What do you mean by high-risk patients?

Lines 49-52: What end organ damage are the authors referring to?

Lines 54-56: Physicians are not needed to perform US. There are technicians who perform the imaging. Physicians simply read the images. Need to cite.

Lines 58: How are miRNAs stable is blood when there are RNAases in blood? They must be associated with proteins or contained in EVs. Please elaborate on this.

Lines 62: Cite papers that have used plasma miRNAs as biomarkers for CV disease. There are several.

Methods

Please define what is meant by high cardiovascular risk.

Please discuss how plasma was isolated from a patient blood sample?

What was the reference gene used for PCR analysis? Most studies use TBP or GAPDH. However, those may not be ideal for this study. I understand that you used spike-ins to evaluate RNA extraction efficiency and rt-qPCR, however, need to provide reference gene/miRNA that the miRNAs of interest were normalized to. 

Change Fold change calculation to 2^-∆∆Ct

Statistics - change media to median. Discuss logistic regression and what variables were included.

When comparing >100 different miRNAs ANOVA is not a sufficient statistical method. It will introduce false positives simply due to the fact that there are so many different comparisons. Need to do a false discovery rate which will tell you the rate of type 1 errors when completing multiple comparisons. Usually for these comparisons, a poisson distribution is employed to calculate maximum likelihood ratios. Suggest discussing with a data scientist.

Results

Line 169 : Move the sentence about controls to methods

Line 171: The authors used 32 patients as control samples. However, they only utilized n=6 for patients with SAD and aneurysm. How was that number decided on?

Line 172-173: Why were some samples (control) pooled while others are not?

Figure 1: Authors don’t need to show the miRNA that had Ct values within thresholds. A statement referring the the miRNAs within thresholds in the methods is sufficient.

Instead would be better to make a list of the top differentially expressed miRNAs between the group and whether they have previously been implicated in aneurysmal disease.

Lines 186-191: The authors evaluated the miRNAs that were differentially expressed in SAD and AAA patients. Thought the goal of the paper was to identify patients with SAD. Why were the AAA patients included in the analysis? I would imagine using the AAA patients as a positive control.

How did the authors decide on the 12 miRNAs? Were these the ones with the great fold-changes or the smallest p-values?

Lines 222-226: Where all these analysis done on an n=6? For the ROCs, where the SAD and AAA groups combined?

Lines 237: Please specify what variables were included in the regression model. This should be in the methods.

Lines 245 - 250: The authors need to discuss why they wanted to correlated miRNAs with clinical parameters.

Table 3: why are p-values for some of these miRNA >0.05? Are these not compared to controls?

Figure 5: How did the authors decide on target genes of miRNA-339-3p? Usually, miRNAs have 100s of targets.

Discussion

The authors need to discuss how a biomarker is helpful in diagnosis of AAA disease. What will this biomarker add? Will it be a screening biomarker? Such that patients don’t need to get ultrasounds? That is unlikely given the low AUCs with the miRNA-339.

The authors should refrain from calling miR-339-3p a “biomarker” given that they had small samples sizes, no validation cohorts, and no external validation. At best this is a discovery cohort.

The authors further discuss that a large cohort of their patients are under the guidelines for AAA screening thus making this “biomarker” useful. However, a 2.5-3 cm aneurysm has a <0.1% risk of rupture. Further, more an ectasia or aneurysm grows 10%/per year on average, thus a 2.5 cm aneurysm will take several years to get to 5.0-5.5 (threshold for repair). Thus, I find it hard to believe that we need a screening tool to find aortic ectasia prior to the current screening guidelines. 

Round 2

Reviewer 2 Report

Authors made all necessary revisions. Paper can be published in the current format. No further edits needed. Congratulations!